# Vaccination Schedule and Age Influence Impaired Responsiveness to Hepatitis B Vaccination: A Randomized Trial in Central Asia

**DOI:** 10.3390/pathogens13121082

**Published:** 2024-12-09

**Authors:** Janyn Heisig, Zuridin Sh. Nurmatov, Peggy Riese, Stephanie Trittel, Gulsunai J. Sattarova, Saikal N. Temirbekova, Gulnara Zh. Zhumagulova, Zhanylai N. Nuridinova, Aisuluu A. Derkenbaeva, Bubuzhan K. Arykbaeva, Bakyt I. Dzhangaziev, Jana Prokein, Norman Klopp, Thomas Illig, Carlos A. Guzmán, Omor T. Kasymov, Manas K. Akmatov, Frank Pessler

**Affiliations:** 1Department Vaccinology and Applied Microbiology, Helmholtz Centre for Infection Research, 38124 Braunschweig, Germany; janynheisig@gmail.com (J.H.); peggy.riese@helmholtz-hzi.de (P.R.); stephanie.trittel@helmholtz-hzi.de (S.T.); carlosalberto.guzman@helmholtz-hzi.de (C.A.G.); 2National Institute of Public Health, Ministry of Health of the Kyrgyz Republic, Bishkek 720005, Kyrgyzstan; zuridin@mail.ru (Z.S.N.); s.gulsun87@mail.ru (G.J.S.); skylychbekova15@gmail.com (S.N.T.); janylay35@gmail.com (Z.N.N.); aisuluu.d.88@gmail.com (A.A.D.); 3Republican Center for Immunoprophylaxis, Ministry of Health of the Kyrgyz Republic, Bishkek 720040, Kyrgyzstan; gulnara.zhumagulova@privivka.kg; 4Ministry of Health of the Kyrgyz Republic, Bishkek 720040, Kyrgyzstan; abk_cgsn@mail.ru (B.K.A.); djangazievb@gmail.com (B.I.D.); 5Hannover Unified Biobank, Hannover Medical School, 30625 Hannover, Germany; prokein.jana@mh-hannover.de (J.P.); klopp.norman@mh-hannover.de (N.K.); illig.thomas@mh-hannover.de (T.I.); 6Scientific and Production Centre for Preventive Medicine, Ministry of Health of the Kyrgyz Republic, Bishkek 720005, Kyrgyzstan; logvinehko@mail.ru; 7Research Group Biomarkers for Infectious Diseases, TWINCORE Centre for Experimental and Clinical Infection Research, 30625 Hannover, Germany; makmatov@zi.de; 8Centre for Individualised Infection Medicine, 30625 Hannover, Germany

**Keywords:** aging, Central Asia, compliance, hepatitis B, immune response, Kyrgyzstan, study retention, vaccine response, vaccination

## Abstract

Vaccination against hepatitis B virus (HBV) is the most cost-efficient measure to prevent infection. Still, vaccination coverage among adults in Central Asia, including Kyrgyzstan, remains suboptimal, and data about immune responses to HBV vaccination are lacking. HBV vaccination is given as three injections, whereby the second and third doses are given 1 and 6 months after the first (0-1-6 scheme). However, compliance with the third dose is low in Kyrgyzstan, presumably due to the long time interval between the second and third doses, suggesting that a shortened vaccination schedule could result in better adherence and increased seroconversion. Thus, we conducted a randomized trial of individuals aged 17–66 years comparing the 0-1-6 scheme against a shorter 0-1-3 scheme. Primary outcome measures were post-vaccination titers and the percentage of participants with protective post-vaccination titers (≥10 mIU/mL). Compliance with the completeness of blood draws and administered third vaccine dose was better with the 0-1-3 scheme than with the 0-1-6 scheme. In both study arms combined, younger age (<40 years) was associated with better vaccine protection. The 0-1-6 scheme resulted in higher post-vaccination titers (52 versus 15 mIU/mL, *p* = 0.002) and a higher seroprotection rate (85% versus 64%, *p* = 0.01) than the 0-1-3 scheme, whereby post-vaccination titers correlated negatively with age in the 0-1-3 scheme. Thus, the 0-1-6 scheme should continue to be the preferred HBV vaccination schedule, but interventions to improve compliance with the third vaccine dose are needed.

## 1. Introduction

Hepatitis B virus (HBV) exerts a global disease burden with high morbidity and mortality. It is estimated that, globally, approximately 290 million people have chronic hepatitis B, and over 800,000 people die each year from infection-related causes [1]. HBV attacks the liver and can cause both acute and chronic disease manifestations, such as cirrhosis, which increases the risk of liver cancer. The prevailing route of HBV transmission is through body fluids or blood from an infected person, which can occur, e.g., through sexual contact or from mother to child at birth [2]. In total, 90% of infants and 30–50% of children younger than five years who are infected in the first years of life will develop a chronic infection. In contrast, only 5% of infected adults will develop chronic manifestations [3,4,5].

Global meta-analyses highlight the continued need for the surveillance, control, and prevention of HBV infection. The worldwide estimate of HBV prevalence is between 2.5% and 5.6% across all ages [3,6,7]. Although progress toward reducing HBV infections has been made, hepatitis B remains a substantial public health problem [3]. Kyrgyzstan is one of the countries targeting the elimination of viral hepatitis. Since its independence from the former Soviet Union in 1991, the country has faced challenges in the health care system and an economic contraction [8,9]. In the last few decades, however, improvements have led to a decline in HBV prevalence from 10% to 2.5–6.6% [3,6,10,11]. For this, a major contribution was the introduction of immunization regimes with the hepatitis B vaccine at birth and their expansion to the whole country [11]. Hence, it was suggested that Kyrgyzstan be reclassified, from a high to a lower-intermediate HBV endemic country [11,12]. However, despite current policies, HBV prevalence among adults continues to be high, with an increasing trend with older age [12,13]. Thus, preventive measures should be established to improve HBV vaccination coverage in this population, as well as for individuals at high risk. There are no studies investigating immune responses to HBV vaccines in Central Asia or factors that might influence vaccination outcomes. However, in clinical practice in the capital, Bishkek, a current challenge is that vaccinees often do not return for the third injection, presumably due to the long time interval between the second and third doses. Since the third dose is essential for the development of full HBV vaccine protection [14,15], this likely contributes to suboptimal HBV vaccine protection in the population. We hypothesized that a shorter time interval between the second and third dose could improve return rates for the third dose. However, before a simplified 0-1-3 scheme could be considered further for vaccine policy, it was critical to assess whether it would result in seroprotection equivalent to the traditional 0-1-6 scheme. Thus, we conducted a randomized trial of individuals aged 17–66 years in which we compared vaccine-conferred seroprotection between the traditional 0-1-6 scheme and a shorter 0-1-3 scheme. In addition, we investigated study protocol compliance and evaluated potential risk factors for an insufficient vaccine response.

## 2. Materials and Methods

### 2.1. Study Design

Participants were recruited from our previously conducted cross-sectional study about viral hepatitis in Kyrgyzstan [13]. In that study, 1073 individuals were screened. HBsAg was detected in sera from 33 individuals (HBsAg seropositive). Of the remaining 1040 individuals, 246 agreed to participate in this study. After completion of the field study, 105 participants were considered for the final analysis based on the availability of serum samples for all four study time points, having received 3 vaccine doses, and being anti-HBsAg seronegative before the 1st vaccine dose. All study visits took place in a family medicine center (FMC) in Bishkek, the capital of Kyrgyzstan. FMCs, formerly known as polyclinics, are responsible for outpatient health care, and there are 19 FMCs in Bishkek. Study visits, including blood draws and vaccination, took place in FMC no. 11, which is responsible for approximately 30,000 residents in Bishkek [13]. The vaccine was administered intramuscularly (deltoid muscle) by nurses trained in the technique.

All study participants received three doses of the inactivated vaccine Regevak-B (Binnofarm Group, Moscow, Russia). The effectiveness of this vaccine had previously been assessed in a field study involving 560 participants in 4 different locations in Russia, wherein 67.2–88.1% of vaccinees developed protective titers after the 3rd dose [16]. Participants in our study were randomized to one of two study arms: one group received the vaccine doses at months 0, 1, and 3 (0-1-3 schedule), whereas the other group received doses at months 0, 1, and 6 (0-1-6 schedule). Blood samples were taken before each vaccination and 19–275 days after the 3rd vaccine dose. Serum samples were obtained by centrifugation and stored at −80 °C at the study site in Bishkek until they were transferred to the Hannover Unified Biobank (Hannover, Germany) after the conclusion of the field study. Subsequently, samples were transferred to the laboratory site at the Helmholtz Centre for Infection Research (Braunschweig, Germany) for determination of anti-HBsAg titers. All samples were continuously kept frozen until the time of serological analyses. The study was approved by the Ethics Committee of the Ministry of Health in Kyrgyzstan (no. 01-2-118). Written informed consent was collected from all participants. Figure 1 illustrates how the final study population was derived from the larger population that was screened in our original cross-sectional study on the prevalence of viral hepatitis in Kyrgyzstan [13].

### 2.2. Serological Assay

Anti-hepatitis B surface antigen (HBsAg) antibody titers were determined using a quantitative anti-HBsAg ELISA kit (Creative Diagnostics, Shirley, NY, USA, cat. no. DEIA060). The assay and calculation of anti-HBsAg titers (mIU/mL) were carried out according to the manufacturer’s protocol. Sample absorbance values (OD_450nm_) that were lower or equal to the Blank OD_450nm_ were assigned a titer of 0.01 mIU/mL to allow for subsequent analysis. Seroprotection was defined by an anti-HBsAg titer of ≥10 mIU/mL. Accordingly, vaccine non-responders were defined as those with an anti-HBsAg titer of <10 mIU/mL.

### 2.3. Statistical Analysis

The significance of differences between the two study groups was determined with a Chi-squared test or a Mann–Whitney U test as indicated. For analysis within a group, the Friedman test was applied. The Spearman correlation coefficient, r, was determined for correlation analyses. All statistical analyses were performed using GraphPad Prism (version 9.4.1, www.graphpad.com).

## 3. Results

### 3.1. Compliance with the Study Protocol

A total of 246 subjects met the inclusion criteria and were recruited into the study. No major adverse events were reported in the study. Compliance with the study protocol was significantly better in the 0-1-3 arm than in the 0-1-6 study arm in terms of completeness of vaccination (three doses, 92.9% versus 61.9%) and blood draws (four blood draws, 53.6% versus 45.5%) (Table 1). Age and sex were not associated with the completeness of vaccinations or blood draws in either of the two schemes.

Nonetheless, the number of participants who received all three vaccine doses and donated all four blood samples was nearly identical (n = 60 [0-1-3] and n = 61 [0-1-6]). In total, 11.5% of the participants in the 0-1-3 vaccination scheme and 15.0% of those in the 0-1-6 scheme turned out to be seropositive for anti-HBsAg antibodies in the baseline blood sample taken just before the first study vaccination and were excluded from further analyses. In total, 105 participants (0-1-3: n = 53; 0-1-6: n = 52) fulfilled the requirements of three vaccine doses, four blood samples, and HBV seronegative status at baseline and were included in the final analysis (Figure 1). The demographics and other characteristics of these two study groups are shown in Table 2. No significant differences were observed in terms of age or sex between the two arms, thereby ruling out any potential contribution of these two factors to emergent differences. Completeness of vaccination (all three doses) was significantly better in the 0-1-3 scheme than in the 0-1-6 scheme, supporting our initial hypothesis that the long time interval between doses 2 and 3 is responsible for the poor return rates of vaccinees in Kyrgyzstan for the third dose. According to the study protocol, post-vaccination titers were to be taken four weeks after the third dose. However, compliance with this aspect of the study protocol was poor, and the median time interval turned out to be 214 days (range, 25–275; study arm, 0-1-3) and 186 days (range, 19–249; study arm, 0-1-6) (*p* = 0.0105).

In addition, data about education level as a socioeconomic factor were available for 90 participants (Appendix A). When stratifying the participants by age (<40 years and ≥40 years), vocational education was significantly higher in participants aged ≥40 years, while uncompleted higher education (university) was significantly more frequent in participants aged below 40 years, but this was due to only two very young participants (Appendix A).

### 3.2. Vaccine-Induced Antibody Responses Among Different Age Groups

We first addressed the question of whether age had an impact on vaccine responses, irrespective of the study arm. In 10-year age groups, most participants were between 20 and 59 years old (88.5%), whereas few participants were younger than 20 years or 60 years or older (Figure 2).

In order to assess the vaccine response, anti-HBsAg antibody levels were measured in serum samples taken before each vaccine dose and once after the third dose. Vaccine-induced geometric mean titers (GMTs) against HBsAg were detected in each study age group from M1 to M4/M7 (Figure 3). After the first vaccination, individuals under 20 years old showed significantly higher GMTs (20.13 mIU/mL) than the other age groups (0.06–1.19 mIU/mL) (Appendix A). The second vaccine dose tended to induce higher GMTs in participants < 40 years (11.05–121.90 mIU/mL) than in participants ≥ 40 years (0.16–1.38 mIU/mL). Following the third vaccine dose, the younger (<40 years) age groups showed significantly higher GMTs (36.70–126.90 mIU/mL) than the older (≥40 years) age groups (5.3–16.74 mIU/mL). No significant differences in GMTs after the third vaccine dose were observed when stratifying the participants according to education level (Appendix A). Overall, study participants < 20 years old exhibited the highest GMTs at any post-vaccination time point. These differences were most pronounced at M1 and M3/M6. Thus, the obtained data emphasize the induction of stronger HBV vaccine responses in younger individuals. However, incremental increases in antibody levels following each vaccine dose were particularly notable in middle age (20–59 years).

### 3.3. Seroprotection Rates Following Vaccination Are Higher in Younger Adults

Furthermore, we addressed the question of whether, according to the Centers for Disease Control and Prevention (CDC), a post-vaccination anti-HBs titer ≥ 10 mIU/mL is associated with immunity to HBV infection [17]. In this regard, the stratification of the study participants revealed an increasing seroprotection rate in each age group following the complete vaccination series (Figure 4A). The majority (60%) in the age group < 20 years developed protective antibody levels after the first dose. In contrast, among the other groups, the protection rate was only 5–19%. After the second vaccine dose, all participants in the age group < 20 years, as well as the majority of participants between 20 and 39 years, developed protective titers (60–71%), whereas seroprotection rates in the older population (≥40 years) were only 14–41%. Following the final vaccine dose, high proportions of individuals with protective levels were found in the age groups < 40 years (81–100%), whereas the seroprotection rates in the older age groups over 40 years remained suboptimal (57–65%). Thus, in accordance with the above-analyzed GMTs, the three-dose vaccination schedule was more effective in younger adults as compared to individuals of older ages. Furthermore, a vaccination series of three doses was necessary for an adequate seroconversion rate. As shown in Figure 4B, the administration of the first and second doses resulted in only 15% and 50% responders, respectively, whereas the third vaccine dose resulted in a total of 74% vaccine responders. Nevertheless, even after completing the full protocol, 26% of the vaccinees were still unprotected.

### 3.4. Similar Vaccine Responses in Female and Male Subjects Regardless of Age

Among all study participants, 65% were females and 35% males. The vaccine-induced anti-HBsAg antibody responses were similar in females and males (Table 3). Following the first vaccine dose, the analysis revealed GMTs of 0.24 mIU/mL for females and 0.18 mIU/mL for males with seroprotection rates of 13% and 19%, respectively. Vaccine-induced GMTs and the percentage of seroprotected individuals increased with the second vaccine dose. Here, females displayed an anti-HBsAg GMT of 3.15 mIU/mL (seroprotection rate, 49%) and males 2.47 mIU/mL (seroprotection rate, 51%). After the last vaccine dose, GMTs of 29.11 mIU/mL for females and 25.48 mIU/mL for males were observed with 72% and 78% of seroprotected subjects, respectively. Overall, the increment in GMTs within a subgroup after each vaccine dose was significant, whereas no significant differences were found between the two sex subgroups at any measured time point. However, 28% of females and 22% of males remained non-responders after completing the vaccination series.

### 3.5. An Insufficient Vaccine Response Is Associated with Shorter Vaccination Schedule and Older Age

We then explored potential differences in vaccine-induced immune responses between the two vaccination schedules (i.e., 0-1-3 versus 0-1-6). As expected, similar GMTs were achieved in the two groups after the first and second vaccinations (Table 4). In contrast, after the third dose, the 0-1-6 scheme resulted in significantly higher anti-HBsAg GMTs and seroprotection rates than the 0-1-3 scheme. Thus, a longer interval between the second and third vaccination induced a more efficient seroresponse.

The observed differences in vaccine responsiveness across the age groups, as well as vaccination schedules, led to a further investigation of associations with those parameters. Anti-HBsAg titers after the third vaccination correlated negatively with age in the 0-1-3 group but not in the 0-1-6 group (Figure 5). Therefore, the obtained results suggest that impaired vaccine responses are linked to a shorter vaccination schedule and increasing age. Furthermore, separate correlation analysis using female and male vaccinees revealed that male sex, increasing age, and the 0-1-3 vaccination schedule were particularly significantly associated with lower antibody titers (Figure 6).

In order to test if the impaired vaccine response with increasing age is linked to education level, a correlation analysis of age and education with anti-HBsAg titers after the third vaccination was performed. No significant correlations were observed (Appendix A).

### 3.6. Effect of the Unusually Long Interval Between Third Vaccine Dose and Titer Determination

Since the envisaged time interval of 30 days between the third dose (M3/M6) and the blood draw for titer determination (M4/M7) could not be achieved, we assessed the impact of differences in this time interval on antibody titers. First, we analyzed all age groups combined. A significant negative correlation was observed between the rise in anti-HBsAg titers (expressed as a fold-change from M3/M6 to M4/M7) and the length of this time interval (Figure 7). This negative correlation persisted when stratifying the participants by age (<40 years and ≥40 years) (Appendix A). Thus, measurements taken at later time points underestimated the true vaccine response independently of age, likely due to a gradual decline in titers after the optimal time point of approximately 30 days.

Next, we tested whether the negative impact of the time interval between the third dose and the time of antibody titer determination could be observed in both vaccination schedules (0-1-3 versus 0-1-6). For this, participants with an interval of >255 days were excluded, thereby assuring a comparable median interval time between the 0-1-3 (n = 47; median days, 200 [95% CI 56–224]) and 0-1-6 (n = 52; median days, 186 [95% CI 123–193]) vaccination groups (*p* = 0.091, Mann-Whitney U test). Anti-HBsAg antibody titers were significantly higher in the 0-1-6 scheme than in the 0-1-3 scheme, and this difference persisted when the participants were further stratified into subgroups with <100 or ≥100 interval days between M3/M6 and M4/M7 (Figure 8). Thus, these results strongly suggest that the classic 0-1-6 vaccination schedule results in stronger antibody responses within 255 days after the third vaccine dose.

## 4. Discussion

Although HBV prevalence has declined in many countries over the last few decades, hepatitis B remains a substantial public health problem and economic burden [18]. Costs for universal health coverage caused by viral hepatitis infections amount to an estimated USD 6 billion per year [19]. The substantial global burden of HBV infections led to the approval of the first global health strategy on viral hepatitis by the World Health Organization (WHO) in 2016. This strategy includes the goal of eliminating viral hepatitis as a major public health threat by 2030 [20]. As a preventive measure, vaccination represents the most cost-efficient tool to reduce the high morbidity and mortality rates of HBV infections since current therapies do not cure the infection. However, the standard hepatitis B vaccine fails to produce a protective immune response in 5% of the general population, with an increasing likelihood of non-response in individuals with chronic conditions and older age [21,22]. Besides this human immunology problem, there are additional challenges that have to be overcome to increase HBV vaccination coverage, including economic and political circumstances in many low- and middle-income countries [23].

In Kyrgyzstan, the monitoring of viral hepatitis prevention remains suboptimal, and data about HBV vaccine responsiveness are lacking. To date, no study exists analyzing the immune responses following vaccination against HBV in Kyrgyzstan. According to previous HBV vaccination studies from other countries, the conventional vaccination schedule (0-1-6 months) induces seroconversion/seroprotection rates of >89% in healthy adults [24,25,26,27,28]. In contrast, our study showed that only 74% of all participants became seroprotective after receiving the full vaccination series. The observed differences compared to other studies are likely explained by the unusually (and unplanned) long time interval between the third dose and the blood draw for the measurement of post-vaccination titers, but differences in study populations (i.e., different median ages or the presence of chronic diseases) and commercial vaccine products may also contribute. It has been reported that age has an impact on immunological responses to HBV vaccines [22,25]. Age-related changes in the immune system, including immune senescence and inflammaging, can lead to impaired immunity in older adults, thereby reducing the prophylactic efficacy of vaccinations as compared to younger individuals [29,30]. In line with this, our study demonstrated that vaccine non-responders were more common among participants ≥ 40 years old, and immunologic processes associated with immune senescence are likely also responsible for the fact that antibody titers correlated negatively with age in the 0-1-3 scheme.

Aiming at a cost-efficient vaccination schedule, previous studies have analyzed accelerated vaccination schedules for HBV vaccines as compared to the standard schedule (0-1-6-month) to provide rapid seroprotection, which might also influence compliance among adults to get vaccinated [24,27,28,31,32,33,34,35,36,37,38]. It has been reported that shorter intervals (<4 months) between the second and third vaccine dose induce a relatively efficient seroresponse (64–92% of participants achieving protective titers; 64% in our study) and may not be inferior to longer intervals [36,37,38]. However, the majority of studies have demonstrated that longer intervals (≥4 months) result in even higher seroconversion rates, as well as anti-HBsAg GMTs, and longer intervals are, therefore, recommended. In accordance with these reports, our study revealed higher vaccine-induced seroprotective immune responses for the 0-1-6-month schedule as compared to the 0-1-3-month schedule after completing the full vaccination series. This can be explained, at least in part, by a putative optimal interval to match the boost with the contraction phase of the germinal center [39]. The kinetics of germinal center contraction have been studied in needle aspirations of human lymph nodes after SARS-CoV-2 vaccination (reviewed in [40]). In analogy to findings in these SARS-CoV-2 studies, it is tempting to speculate that a better match with the germinal center contraction in the 0-1-6 scheme results in the improved recruitment of memory B cells that re-encounter the antigen and re-enter the GC. This would lead to improved hypersomatic mutation and affinity maturation, which would improve the quality of plasmablasts and memory B cells. This would result not only in higher titers but also in a better quality of antibodies in terms of affinity, avidity, and neutralization capacity. Applied to our study, this model rests on the assumption that germinal center contraction after HBsAg vaccination follows different kinetics with advancing age. Future studies should be directed at testing this hypothesis. It is also generally assumed that higher anti-HBsAg levels after the primary vaccination series result in longer-lasting vaccine protection [41]. In this context, previous studies have demonstrated that women develop higher antibody levels after vaccination against HBV than men and that vaccine non-responsiveness is more likely among males [42,43,44]. In our study, no sex-specific differences were observed in vaccine-induced antibody responses and seroprotection rates, irrespective of age. However, the correlation analysis between age and sex with the anti-HBsAg titers after the third vaccination within the 0-1-3-month schedule revealed a negative correlation of increasing age with the antibody titers among male adults. Therefore, it is recommended to use a longer schedule for future vaccination strategies, in particular for older men.

However, the present study holds some limitations. We did not evaluate other modifiable risk factors (e.g., psychosocial factors such as beliefs and attitudes regarding vaccines, socioeconomic factors such as occupation status, and education level) for noncompliance with the third vaccine dose. The size of the cohort was relatively small, which also led to unequally distributed age and sex subgroups. Furthermore, information about biomedically important parameters, such as smoking, body mass index, or co-morbidities, was not assessed in this study. These parameters might influence outcomes in terms of responsiveness to vaccination, as well as weaken conclusions on putative differences between the two subcohorts in case of an uneven distribution [42]. Therefore, larger and more in-depth characterized cohorts are needed to confirm our findings. It would also be important to identify incentive strategies to improve participants’ compliance with a timely return for the blood draw after the third vaccine dose.

In conclusion, the results of this regional study in Kyrgyzstan suggest that the conventional 0-1-6-month vaccination schedule in adults provides better seroprotection than the accelerated 0-1-3-month schedule, particularly for older individuals, while compliance with presenting for the third vaccine dose is superior in the 0-1-3-month schedule.

To overcome this bottleneck, improved vaccination strategies, including incentives to improve return rates for the third vaccine dose, specifically tailored for the elderly and the local psycho-socio-economic environment, are urgently needed.

## Figures and Tables

**Figure 1 pathogens-13-01082-f001:**
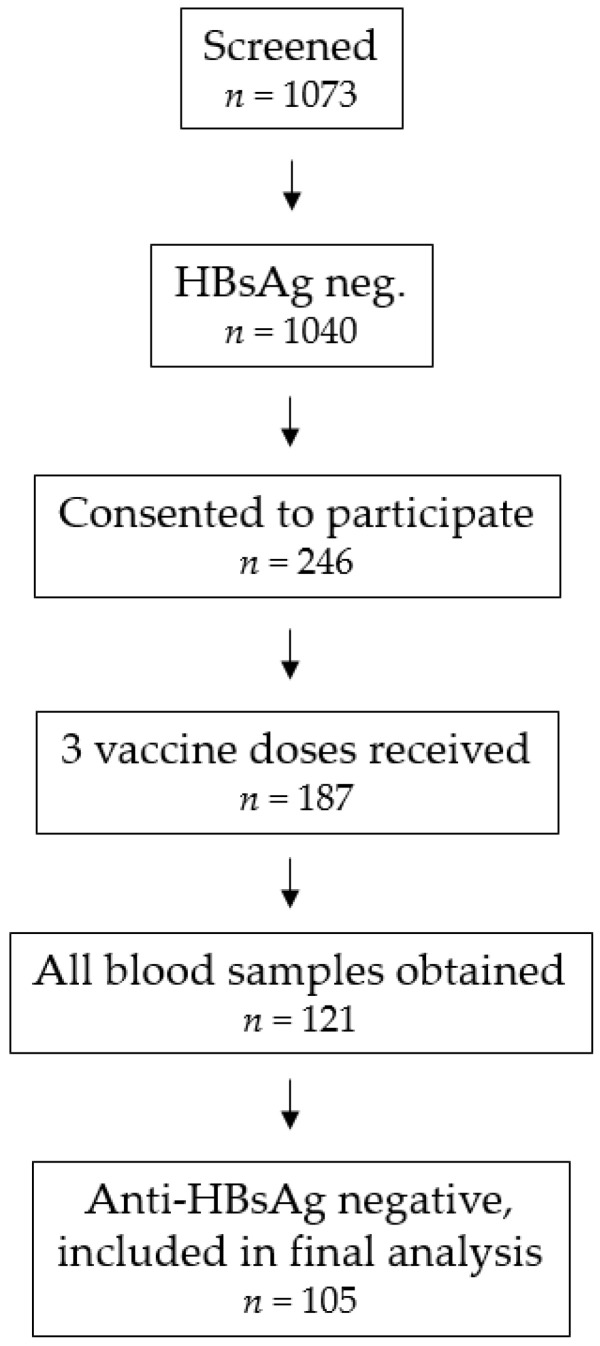
Flow chart illustrating how the final study population was derived from the larger population that was screened in the original cross-sectional study on the prevalence of viral hepatitis in Kyrgyzstan [13].

**Figure 2 pathogens-13-01082-f002:**
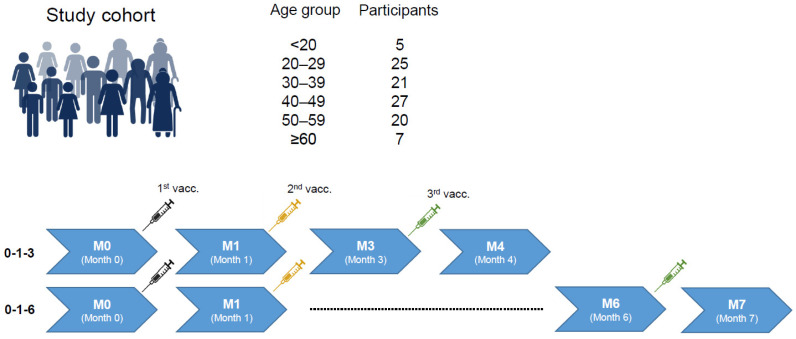
Study design and age distribution. Participants received the 3-dose vaccination series against HBV at months 0, 1, and 3 or 6. Blood was drawn directly before each vaccination and once after the 3rd vaccination. Created with BioRender.com.

**Figure 3 pathogens-13-01082-f003:**
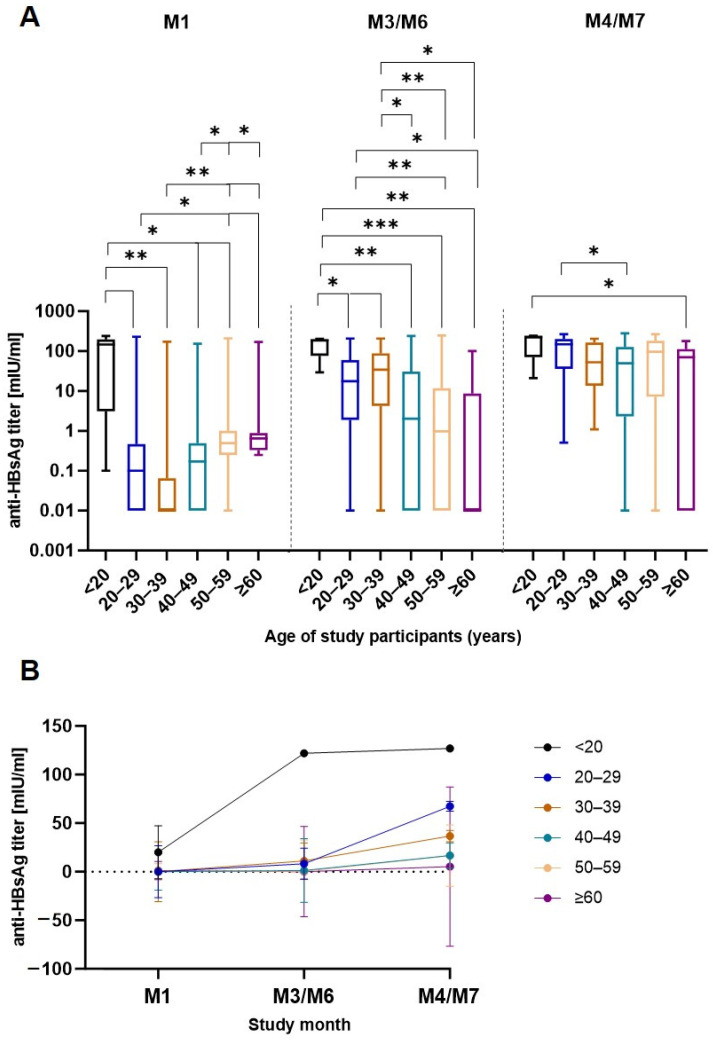
Anti-HBsAg antibody responses after each vaccine dose. (**A**) Box plots depict anti-HBsAg titers (minimum to maximum) with median and quartiles for each age group. (**B**) The same data are represented as line graphs with GMTs and SD. Mann–Whitney U test. * *p* ≤ 0.05; ** *p* ≤ 0.01; *** *p* ≤ 0.001.

**Figure 4 pathogens-13-01082-f004:**
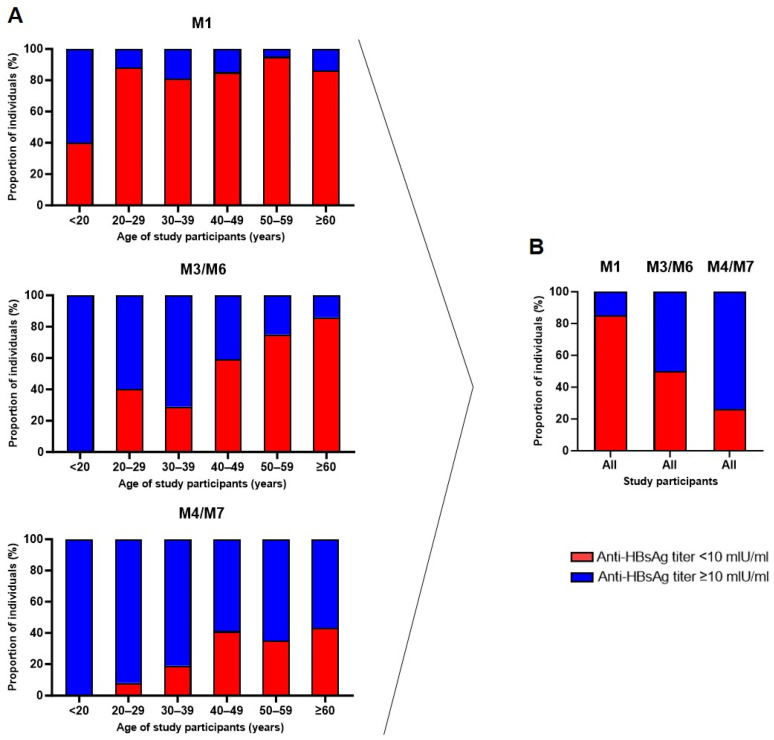
Seroprotection rates and vaccine (non-)responders. (**A**) Proportion of individuals in each study age group according to seroprotection status (not seroprotected = anti-HBsAg level < 10 mIU/mL; seroprotected = anti-HBsAg level ≥ 10 mIU/mL). (**B**) Stratification of all participants combined.

**Figure 5 pathogens-13-01082-f005:**
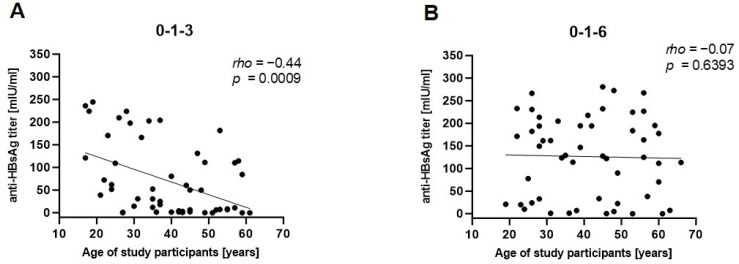
Correlation analysis of age and anti-HBsAg titers after the 3rd vaccination. Associations between the age of study participants and antibody titers after the 3rd vaccination within either the (**A**) 0-1-3- or (**B**) 0-1-6-month vaccination schedule were assessed by Spearman correlation.

**Figure 6 pathogens-13-01082-f006:**
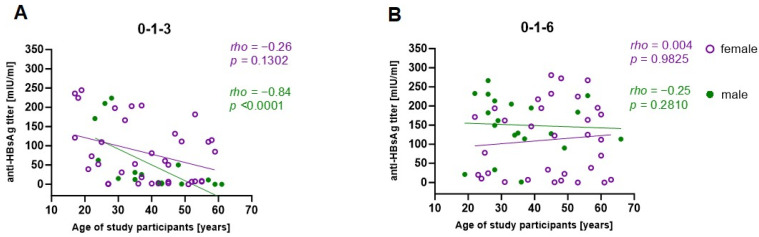
Correlation analysis of age and sex with anti-HBsAg titers after the 3rd vaccination. Associations between the age and sex of study participants with antibody titers after the 3rd vaccination within either the (**A**) 0-1-3- or (**B**) 0-1-6-month vaccination schedule were performed with the Spearman correlation test. Female (purple). Male (green).

**Figure 7 pathogens-13-01082-f007:**
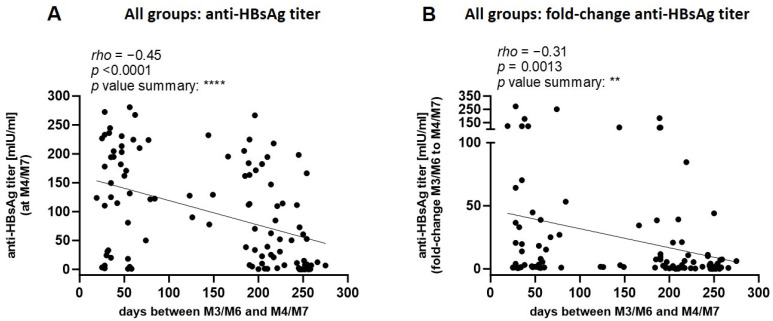
Spearman correlation analysis of anti-HBsAg titers, fold-change, and interval days from M3/M6 to M4/M7. Graphs depict data from all participants combined. (**A**) Correlation between anti-HBsAg titers at M4/M7 and interval days. (**B**) Correlation between fold-change of anti-HBsAg titers from M3/M6 to M4/M7 and interval days. The fold-change was computed by the ratio of the anti-HBsAg titer measured at M4/M7 to the titer measured at M3/M6. ** *p* ≤ 0.01; **** *p* ≤ 0.0001.

**Figure 8 pathogens-13-01082-f008:**
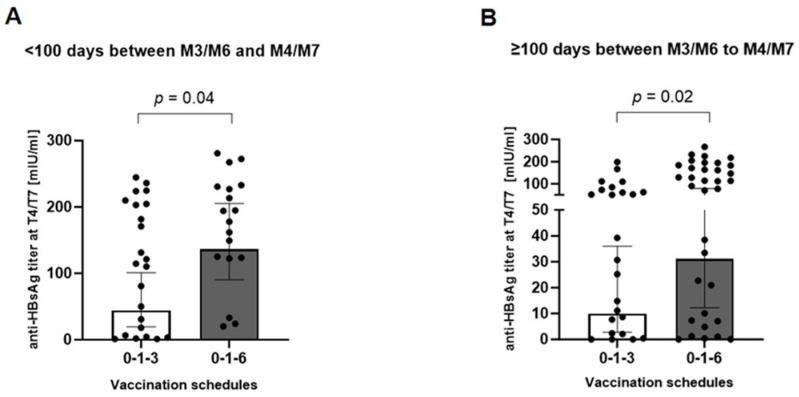
Impact of the interval time from M3/M6 to M4/M7 on anti-HBsAg titers between the vaccination schedules. Participants with interval days of (**A**) <100 and (**B**) 100–255 from M3/M6 to M4/M7 were stratified according to their vaccination schedule (0-1-3 or 0-1-6). Significance of between-group differences was assessed by Mann–Whitney U test.

**Table 1 pathogens-13-01082-t001:** Compliance with the study protocol.

	Initially Randomized to Study Arm	
	0-1-3 (n = 112)	0-1-6 (n = 134)
Completeness of vaccinations		
3 doses	92.9%	61.9%
2 doses	7.1%	32.1%
1 dose	0%	6.0%
Completeness of blood draws		
4 blood draws	53.6%	45.5%
3 blood draws	43.8%	32.8%
2 blood draws	2.7%	15.7%
1 blood draw	0%	6.0%

**Table 2 pathogens-13-01082-t002:** Characteristics of the final study population.

	All	0-1-3	0-1-6	*p* Value
Participants, N	105	53	52	n/a
Median age in years (range)	40 (17–66)	40 (17–61)	41.5 (19–66)	0.3769 ^a^
Female sex, n (%)	68 (64.8%)	36 (67.9%)	32 (61.5%)	0.5437 ^b^
Median time between 3rd dose and 4th blood draw in days (range)	190 (19–275)	214 (25–275)	185.5 (19–249)	0.0105 ^a^

^a^ Mann–Whitney or ^b^ Chi-squared test was applied for differences between two vaccination schedules. n/a = not applicable. Appendix A shows the median age in 10-year increments.

**Table 3 pathogens-13-01082-t003:** Anti-HBsAg GMTs and seroprotection rates of vaccinees stratified according to sex. Data are presented as GMTs with 95% CI.

Sex	M1	M3/M6	M4/M7	*p* Value
Anti-HBsAg GMTs (95% CI)				
Female	0.24 (0.12, 0.50)	3.15 (1.4, 6.70)	29.11 (16.08, 52.69)	<0.0001 ^a^
Male	0.18 (0.06, 0.59)	2.47 (0.69, 8.84)	25.48 (9.65, 67.29)	<0.0001 ^a^
*p*-value	0.3492 ^b^	1.0 ^b^	0.7524 ^b^	0.0105 ^a^
Seroprotection rate, n (%)				
Female	9/68 (13%)	33/68 (49%)	49/68 (72%)	
Male	7/37 (19%)	19/37 (51%)	29/37 (78%)	
*p*-value	0.439 ^c^	0.782 ^c^	0.479 ^c^	

^a^ Friedman test was applied for the comparison of GMTs within one sex. ^b^ Mann–Whitney test or ^c^ Chi-squared test was applied for between-sex comparisons.

**Table 4 pathogens-13-01082-t004:** Comparison between the 0-1-3 and 0-1-6 vaccination schedules.

Vaccination Scheme	M1	*p*-Value	M3/M6	*p*-Value	M4/M7	*p*-Value
Anti-HBsAg GMTs (95% CI)						
0-1-3	0.17 (0.07, 0.39)	0.4242 ^a^	2.02 (0.76, 5.40)	0.323 ^a^	15.05 (7.03, 32.24)	0.0016 ^a^
0-1-6	0.28 (0.11, 0.72)		4.17 (1.63, 10.64)		51.86 (27.03, 98.54)	
Seroprotection rate, n (%)						
0-1-3	7/53 (13.2%)	0.56 ^b^	23/53 (43.4%)	0.21 ^b^	34/53 (64.2%)	0.01 ^b^
0-1-6	9/52 (17.3%)		29/52 (55.8%)		44/52 (84.6%)	

^a^ Mann–Whitney test and ^b^ Chi-squared test were applied for differences between GMTs and seroprotection rates of the two vaccination schedules, respectively.

## Data Availability

The underlying data are accessible at https://figshare.com/search?q=Frank+Pessler&sortBy=posted_date&sortType=desc (accessed on 4 December 2024).

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
