# Peer review of "Vaccination Schedule and Age Influence Impaired Responsiveness to Hepatitis B Vaccination: A Randomized Trial in Central Asia"

_pathogens, 2024, doi:10.3390/pathogens13121082_

Round 1

Reviewer 1 Report

Comments and Suggestions for Authors

The manuscript by Heisig et al. described a field study of two schedules for HBV vaccination in a resource poor country, where elderly people fail to show up after 5 months from the previous vaccine shot; however the results indicate that the increased  compliance is obtained at the expense of a protective antibody response. The message is clear and the study deserves publication. However I have two points that deserve clarification: since the non responder rate seems to be very high, I would like to know of other studies with the Russian vaccine used in other countries and how they compare; second, the better response and compliance in youger people might also be due to socioeconomic factors (e.g. living in cities vs countryside, having more instruction, having higher income) which should be analysed.

Author Response

However I have two points that deserve clarification: since the non responder rate seems to be very high, I would like to know of other studies with the Russian vaccine used in other countries and how they compare;

-- We have added text in Methods about the performance of this vaccine in another "real-life" study, which was conducted in Russia. Indeed, seroprotection varied according to the study region and is in the range of the seroprotection observed by us. 

second, the better response and compliance in youger people might also be due to socioeconomic factors (e.g. living in cities vs countryside, having more instruction, having higher income) which should be analysed.

-- All participants hailed from the region including the capital. Thus, there were no participants from a strictly rural environment. We had collected data about education level, but not on income. Stratification according to education level did not reveal any association between education level and seroprotection. We have added these findings to the Results section and to the Supplement. 

Reviewer 2 Report

Comments and Suggestions for Authors

In the manuscript submitted to me for review entitled "Vaccination schedule and age influence impaired responsiveness to hepatitis B vaccination: a randomized trial in Central Asia“ the authors Janyn Heisig, Zuridin Sh. Nurmatov, Peggy Riese, Stephanie Trittel, Gulsunai J. Sattarova, Saikal N. Temirbekova, Gulnara Zh. Zhumagulova, Zhanylai N. Nuridinova, Aisuluu Derkenbayeva, Bubuzhan K. Arykbaeva, Bakyt I. Dzhangaziev, Jana Prokein, Norman Klopp, Thomas Illig, Carlos Alberto Guzman, Omor T. Kasymov, Manas K. Akmatov and Frank Pessler present a study in which they followed the effect of the administration of a shorter vaccination scheme against hepatitis B virus (HBV). They follow the defensive ones titers after vaccination and compared them with those of the standard 0 - 1 - 6 vaccination scheme, in different age groups and in both sexes.

Individuals aged 17 - 66 years were included in the study. The study was approved by the Ethics Committee of the Ministry of Health in Kyrgyzstan and was in accordance with national legislation and the Declaration of Helsinki. All study participants gave prior written consent to be included in the study.

My remarks and recommendations to the authors are:

1. Why was the minimum age limit of 17 years selected? Why not 18 for example?

2. The same omission is made in several places in the text. It is indicated that certain results are tabulated, but the table number is not given. I have noticed a similar omission on lines 178, 187, 269, 273, 295 and 316. Let the number of the table in which the results are presented be indicated in all places.

3. It is clear from the study that the new proposed scheme of vaccination 0 -1 -3 does not give better results than the established scheme 0 - 1 - 6. The only advantage of the scheme is the greater percentage of patients who received the third dose of vaccination.

It is not a required part of the structure of the manuscript, but I think it would be good for the readers if it is stated in a short Conclusion, although it is commented on in the results and discussion of the manuscript.

4. Since the immune response to vaccination is better in younger individuals, are there any data on the use of the HBV vaccine in children?

Author Response

  1. Why was the minimum age limit of 17 years selected? Why not 18 for example?

--- There was no scientific reason for setting the cut-off at 17 years. 

2. The same omission is made in several places in the text. It is indicated that certain results are tabulated, but the table number is not given. I have noticed a similar omission on lines 178, 187, 269, 273, 295 and 316. Let the number of the table in which the results are presented be indicated in all places.

--- We have corrected this formatting error, which was due to the use of digital links to figures and tables. 

3. It is clear from the study that the new proposed scheme of vaccination 0 -1 -3 does not give better results than the established scheme 0 - 1 - 6. The only advantage of the scheme is the greater percentage of patients who received the third dose of vaccination.

It is not a required part of the structure of the manuscript, but I think it would be good for the readers if it is stated in a short Conclusion, although it is commented on in the results and discussion of the manuscript.

--- We added a sentence to Discussion that the 0-1-3 scheme did result in better compliance with presenting for the 3rd vaccine dose. 

  1. Since the immune response to vaccination is better in younger individuals, are there any data on the use of the HBV vaccine in children?

--- We could not find any data on efficacy of the vaccine used in our study in a pediatric population. The publication by Kozhanova et al. (new reference 16 in our manuscript) only stratifies participants into younger and older than 40 y.